# Modeling Analysis on Coupling Mechanisms of Mountain–Basin Human–Land Systems: Take Yuxi City as an Example

Li Wu [1,2], Yanjun Yang [1] and Binggeng Xie [2,*]

1 Department of Geography and Land Engineering, Yuxi Normal University, Yuxi 653100, China; wuli2009@yxnu.edu.cn (L.W.); yangyanjun@yxnu.edu.cn (Y.Y.)
2 School of Geographic Sciences, Hunan Normal University, Changsha 410081, China
* Correspondence: xbgyb1961@163.com; Tel.: +86-138-0847-3925

**Abstract:** The result of a human–land relationship in geographical environment systems is a human–land coupling system, which is a comprehensive process of interaction and infiltration between human economic and social systems and the natural ecosystem. Based on the recognition that the human–land system is a nonlinear system coupled by multiple factors, a time delay fractional order dynamics model with a Holling-II-type transformation rate was constructed, the stability analysis of the system was carried out, the transformation times of different land classes were clarified, and the coupled dynamics model parameters of mountainous areas and basin areas were obtained by using the land-use change survey data and socio-economic statistical data in Yuxi City, respectively: the transformation parameter of the production and living land to the unused land in mountainous areas and basin areas ($a_M$, 0.0486 and $a_B$, 0.0126); the transformation parameter of unused land to production and living land in mountainous areas and basin areas ($b_M$ 0.0062 and $b_B$, 0.0139); the transformation parameter of unused land to the forest and grass land in mountainous areas and basin areas ($s_M$, 0.0051 and $s_B$, 0.0028); the land area required to maintain the individual unit in mountainous areas and basin areas ($h_M$, 0.0335 and $h_B$, 0.0165); the average reclamation capacity in mountainous areas and basin areas ($d_M$, 0.03 and $d_B$, 0.05); the inherent growth rate of populations in mountainous areas and basin areas ($r_M$, 0.0563 and $r_B$, 0.151). Through analyzing the coupling mechanisms of human–land systems, the countermeasures for the difference between mountainous areas and basin areas in the future development are put forward. The mountainous area should reduce the conversion of forest and grass land to production and living land by reducing the average reclamation or development capacity, reducing the excessive interference of human beings on unused land, and speeding up its natural recovery and succession to forest and grass land. In addition to reducing the average reclamation or development capacity in basin areas, the reclamation or development rate of the idle land and degraded land should be increased, and the conversion of idle land and degraded land into productive and living land should be encouraged by certain scientific and technological means.

**Keywords:** mountain–basin human–land system; land-use change; land dynamical model with Holling-II type; coupling mechanism; Yuxi City

## 1. Introduction

Human–land relationship research is of great significance in geography, contributing to the duality of geography and the development of human geography [1]. For a long time, the study of "humans" and "land" has been carried out separately. However, as a series of environmental problems and food security problems brought about by human activities on the earth continue to affect the human system [2–4], the academic circle is paying more and more attention to the comprehensive research of "people" and "land" [5]. A variety of new comprehensive methods, including statistical methods, GIS and spatial

analysis methods, simulation methods and hybrid methods have been applied [6–9]. As the questions raised by researchers increasingly involved the intersection of human activities and the earth's environmental system [10,11], the academic circle further recognized that the modeling of feedback between humans and the natural environment has become an urgent requirement [12,13]. In the 1980s, the modeling concept of coupled natural systems and human socio-economic systems was proposed, and the two-way coupling of positive and negative feedback and the coupling with human activities in the earth system model became the research object of academic community [14]. Synthetic integrated models that carried out bidirectional coupling and exchanges of information in certain forms have increasingly become a research hotspot [15]. Since the 1990s, 11 different Integrated Assessment Models (IAMs) have been developed worldwide [16]. The Integrated Model to Assess the Global Environment Framework (IMAGE) model developed by the Netherlands Environmental Assessment Agency is one of the representative models of comprehensive integration, in which the impacts of agricultural land expansion and changes in land-use types on the environment were evaluated by considering population density, resources, topography, etc. [17]. In addition, there are some models based on multi-agents [18] that analyze and explain the complex human–land coupling relationship and its coupling degree. Meanwhile, with the continuous development of computer technology, multi-source data-model fusion has made new progress, and the uncertainty of the human–land system coupling relationship has been further quantified [19,20].

With the deepening of studies on the human–land relationship, regional spatiality has attracted more and more attention [21,22], but most relevant studies on this complex issue focus on a single factor [23]. Mountainous areas and basin areas, as special geomorphic spaces in Yunnan Province, have not been strictly subdivided in existing studies, and the relationship between humans and land is rarely involved. The concept of "coupling" in geography originated from physics, which refers to the synergy of two or more systems through various interactions, or the dynamic relationship between the elements of the system [24]. Mountainous areas and basin areas mainly include flat land between mountains and surrounding mountains [25]. The two have a close genetic relationship in topography and geomorphology. Relying on their geographical proximity, they form a complex coupling system of mutual cooperation and constraints through continuous material circulation, energy flow and information transmission, including the two coupling relationships of near-range coupling and remote coupling [11,26,27]. In order to deeply reveal the interactions and feedback mechanisms between human activities and the natural environment in the mountain–basin human–land coupled system, it is necessary to conduct coupling simulations and predictions around the human–land system and build a comprehensive integrated human–land system dynamics model. By analyzing the interaction of element coupling and process coupling between two different geographic spaces, the complexity and dynamics of human–land systems coupling are revealed, and the mechanism and feedback paths of human activities such as social and economic development on land-use changes are explored. The human–land coupling system for mountain–basin has the nonlinear dynamics and chaotic characteristics of complex systems. To obtain a quantitative expression in the structure and function process, it is necessary to refer to a mature paradigm of the existing research and actively explore more integrated multi-variate coupling models to dynamically resolve the interaction coupling relationship and dynamics mechanisms within the complex system and among subsystems based on an interdisciplinary perspective. An outstanding feature of human–land system dynamics models is that it can deal with nonlinear, complex, long-term and dynamic system coupling problems, and it is one of the main models to simulate human–land systems and other complex giant systems [28].

In 1997, Dobson published "Hopes for the Future: Restoration Ecology and Conservation Biology" in the journal Science [29], and proposed a dynamic land model to describe the transformation and restoration of natural habitats, which can explain the driving mechanism of increasing populations' agricultural demand on natural habitat transformation.

However, when a mathematical model needs to be established to solve many specific problems in reality, the time delay cannot be ignored, and it is also one of the essential characteristics of the evolution and interaction results of the human–land systems' elements. From the point of the dynamic system, the existence of a time delay can induce the stability of the system to switch, resulting in complex dynamic behaviors such as periodic oscillation and chaos. Therefore, it is quite necessary to consider the dynamic properties of the land dynamics model with a time delay [30]. In addition, fractional order calculus is an arbitrary generalization of integer order calculus in order, and calculus is widely used in the study of complex dynamic systems, such as the regulation of various ecosystems [31,32], secure communication [33,34], system controls [35,36] and stability issues [37]. Compared with the classical integer order model, fractional order calculus is more suitable for describing systems or processes with memory and hereditary characteristics, and can more accurately describe the physical and ecological phenomena in nature [38,39], which has attracted great attention from scholars at home and abroad [40–43].

Based on this, according to the relatively closed mountain–basin human–land system in Yuxi City, this study took advantage of the limitations on population density and introduced an appropriate land-use conversion rate to focus on analyzing the differences in land-use conversion and population changes over time in two different geographical spaces. On this basis of the land dynamics model and fractional calculus theory constructed by Dobson, a fractional human–land coupling dynamics model with a time delay was established to analyze the evolution mechanism of regional land-use systems and other issues, which is helpful and has important theoretical significance and a practical application value for the in-depth interpretation of the land-use system change mechanism with population development. It also provides reference for the differential human–land countermeasures of mountainous areas and basin areas in different development stages.

## 2. Materials and Methods

### 2.1. Study Area

Yuxi City is located in the central part of Yunnan Province, on the Yunnan Plateau at low latitudes. It belongs to the subtropical plateau monsoon climate, ranging from 23°19′ to 24°53′ north latitude and 101°16′ to 103°09′ east longitude (Figure 1). Yuxi is located in the core position of Yunnan Province, connecting the east to the west and connecting the north to the south. It is adjacent to the provincial capital, Kunming, which is to the northeast; Chuxiong Autonomous Prefecture in the north; Pu'er city in the southwest and Honghe Autonomous Prefecture in the southeast. The city covers an area of 15,285 km$^2$ and has jurisdiction over 75 townships (towns and streets) in 7 counties and 2 districts [44]. The terrain of Yuxi City is high in the northwest and low in the southeast. The western part is mainly deep-cut alpine and valley landforms, the central and eastern parts belong to the mountainous areas of central Yunnan and are dominated by mid-mountain landforms, and the eastern part is mainly plateau lake basin landforms. The Chengjiang, Jiangchuan and Tonghai lacustrine basins are formed around three plateau rifted lakes, the Fuxian Lake, Xingyun Lake and Qilu Lake, with flat and open terrain [45]. According to its special topography, combined with administrative regions, it can be divided into two types of geographical spaces: mountainous areas and basin areas [44]. Due to the complex terrain and large height difference, the mountainous area generally has more rainfall than the basin area. The cultivated land in the mountainous area is shallow and the soil fertility is low, but the basin area has fertile soil and more farmland with high and stable yields. From 1995 to 2018, the urban population growth and economic development in the basin area were significantly higher than those in the mountainous area, and the land-use change and social and economic development status differed significantly between the mountainous area and the basin area [27].

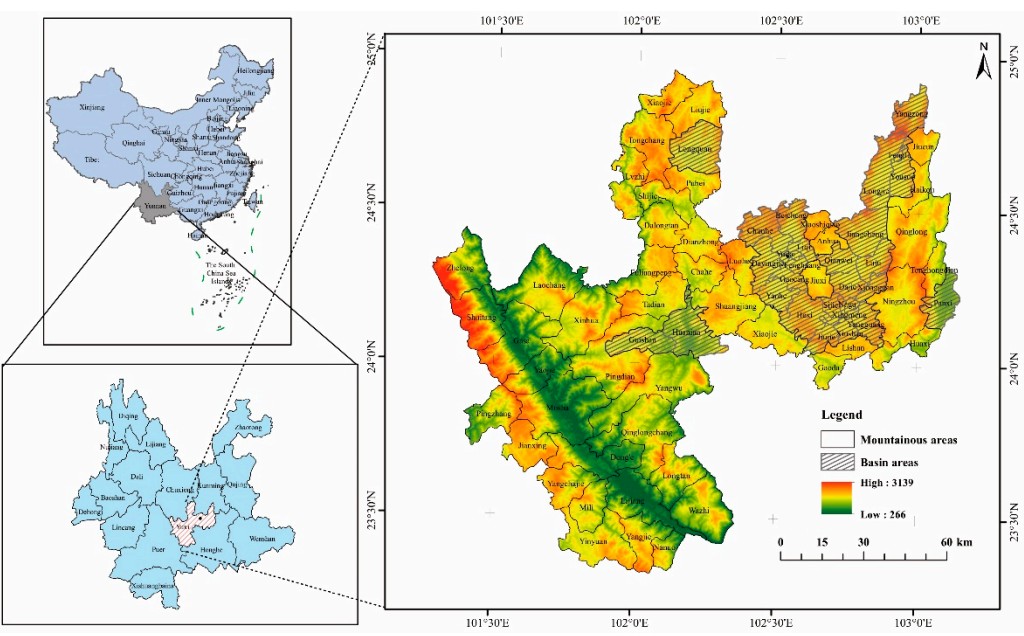

**Figure 1.** Location and elevation of the study area.

*2.2. Data Sources*

The land-use survey data in this study are mainly from the annual change survey data based on the second national land-use survey data (Table 1). The social and economic data involved are mainly from the statistical yearbook of Yunnan Province (1996–2019), the statistical yearbook of Yuxi City (1995–2018), the statistical yearbook of all counties and districts of Yuxi City and the statistical bulletin of national economic and social development from 1995–2018 (Table 1). The role of these data in the research is mainly to train and fit the parameters of human–land coupling dynamics models based on long time series data.

**Table 1.** Land-type area and population changes in mountainous and basin areas of Yuxi City from 1995 to 2018 (unit: hm², person).

| Year | Mountainous Areas | | | | Basin Areas | | | |
|---|---|---|---|---|---|---|---|---|
| | Forest and Grass Land | Production and Living Land | Unused Land | Population | Forest and Grass Land | Production and Living Land | Unused Land | Population |
| 1995 | 875,820.94 | 207,305.42 | 85,826.37 | 931,088 | 179,807.93 | 93,783.09 | 51,990.05 | 974,706 |
| 1996 | 874,606.31 | 208,733.90 | 85,612.52 | 937,439 | 179,726.11 | 93,889.87 | 51,965.09 | 988,002 |
| 1997 | 873,145.48 | 210,799.69 | 85,007.56 | 944,984 | 179,675.47 | 93,964.25 | 51,941.36 | 1,001,365 |
| 1998 | 871,670.98 | 212,451.72 | 84,830.03 | 953,568 | 179,675.23 | 93,969.09 | 51,936.76 | 1,017,639 |
| 1999 | 871,436.37 | 213,042.87 | 84,473.49 | 961,434 | 179,625.42 | 94,070.12 | 51,885.53 | 1,029,912 |
| 2000 | 871,005.39 | 213,891.79 | 84,055.56 | 972,572 | 179,933.94 | 93,832.88 | 51,814.25 | 1,044,208 |
| 2001 | 872,014.87 | 212,901.29 | 84,036.57 | 979,242 | 180,761.39 | 93,006.43 | 51,813.26 | 1,054,823 |
| 2002 | 873,424.16 | 211,633.56 | 83,895.01 | 987,818 | 180,841.31 | 93,172.41 | 51,567.35 | 1,066,139 |
| 2003 | 875,598.05 | 210,385.32 | 82,969.37 | 990,615 | 181,235.14 | 92,977.79 | 51,368.14 | 1,077,858 |
| 2004 | 876,178.05 | 210,009.47 | 82,765.21 | 994,502 | 181,505.02 | 92,855.50 | 51,220.55 | 1,091,030 |
| 2005 | 875,941.02 | 210,234.29 | 82,777.43 | 993,708 | 181,290.67 | 93,097.38 | 51,193.02 | 1,097,941 |
| 2006 | 875,921.89 | 210,243.63 | 82,787.22 | 997,151 | 181,477.04 | 93,131.64 | 50,972.39 | 1,109,013 |
| 2007 | 876,001.57 | 209,665.68 | 83,285.49 | 1,002,592 | 182,061.55 | 92,746.59 | 50,772.93 | 1,119,930 |
| 2008 | 876,136.67 | 209,528.51 | 83,287.55 | 1,001,682 | 181,973.36 | 92,988.61 | 50,619.10 | 1,128,072 |
| 2009 | 876,243.64 | 209,477.35 | 83,231.74 | 1,006,236 | 180,517.24 | 95,251.30 | 49,812.53 | 1,137,356 |

**Table 1.** *Cont.*

| Year | Mountainous Areas | | | | Basin Areas | | | |
|---|---|---|---|---|---|---|---|---|
| | Forest and Grass Land | Production and Living Land | Unused Land | Population | Forest and Grass Land | Production and Living Land | Unused Land | Population |
| 2010 | 876,416.81 | 209,309.41 | 83,226.51 | 1,006,097 | 179,974.29 | 95,977.57 | 49,629.21 | 1,139,411 |
| 2011 | 876,575.39 | 209,073.85 | 83,303.49 | 1,010,814 | 179,666.23 | 96,396.98 | 49,517.86 | 1,148,713 |
| 2012 | 877,034.45 | 208,619.57 | 83,298.71 | 1,010,315 | 179,505.91 | 96,597.58 | 49,477.58 | 1,155,451 |
| 2013 | 877,516.46 | 208,037.60 | 83,398.67 | 1,010,294 | 179,357.88 | 96,787.15 | 49,436.04 | 1,162,075 |
| 2014 | 877,662.39 | 207,725.11 | 83,565.23 | 1,016,317 | 179,146.49 | 97,031.90 | 49,402.68 | 1,168,698 |
| 2015 | 877,862.18 | 207,073.35 | 84,017.20 | 1,014,409 | 178,926.28 | 97,270.57 | 49,384.22 | 1,170,915 |
| 2016 | 878,148.01 | 206,650.20 | 84,154.52 | 1,018,955 | 178,707.29 | 97,537.97 | 49,335.81 | 1,181,352 |
| 2017 | 878,526.13 | 206,159.85 | 84,266.75 | 1,022,613 | 178,519.97 | 97,851.60 | 49,209.50 | 1,192,978 |
| 2018 | 879,014.29 | 205,543.20 | 84,395.24 | 1,025,623 | 178,183.32 | 98,277.02 | 49,120.73 | 1,202,647 |

### 2.3. Human–Land Coupling Model Construction

When discussing land-type transformation, Dobson only considered the direct transformation from the natural habitat to the agricultural land, but did not consider the direct transformation from the natural habitat to the construction land. According to the status of land-use changes in Yuxi City, this study has different definitions of land types based on the original model. Through the combination of land-use types in Yuxi City, it can be divided into the following three types: (1) Forest and grass land: they mainly represent the natural habitat and are set as the original state of land. The forest and grass land in this study are mainly the combination of forest and grass land. (2) Land for production and living: both the agricultural land and construction land transformed from the natural habitat under the current situation are taken into account. Therefore, the farmland, construction land and other necessary land for production and living are combined and collectively referred to as the production and living land. (3) Unused land: the land that cannot be used temporarily due to bad conditions, or after artificial reclamation or productive and living utilization, or long-term unmanaged and barren land. Let the area of forest and grass land in Yuxi be $F$, the production and living land be $R$, and the unused land be $U$, and $N = F + R + U = 1$. At the same time, the following assumptions are made:

a. The evolution of land use begins with the forest and grass land. The area of the forest and grass land at time $t$ is $F(t)$. In order to maintain the survival of human beings, the forest and grass land needs to be reclaimed or developed and converted into production and living land (farmland or construction land). The area of the production and living land at time $t$ is $R(t)$, and it is assumed that the land reclamation or development rate is related to both the individual reclamation or development capacity $d$ and population density $P(t)$.

b. The unused land area at time $t$ is $U(t)$. Assuming that the transformation rate of unused land $b$ is only related to humans' ability to transform the land through science and technology, then $b$ is controllable, that is, adjustable. Let $h$ be the land area required to sustain a single individual.

c. Since there will be a time delay in the evolution of land use, the transfer mechanism of three land types is shown in Figure 2. The area of production and living land at time $t$ is $R(t)$. Due to the abandoned farmland or construction land, it can be converted into unused land $U(t)$ within a period of $1/a$, and become forest and grass land through natural succession or ecological restoration after a period of $1/s$. The unused land can also be converted into production and living land after a time interval of $1/b$.

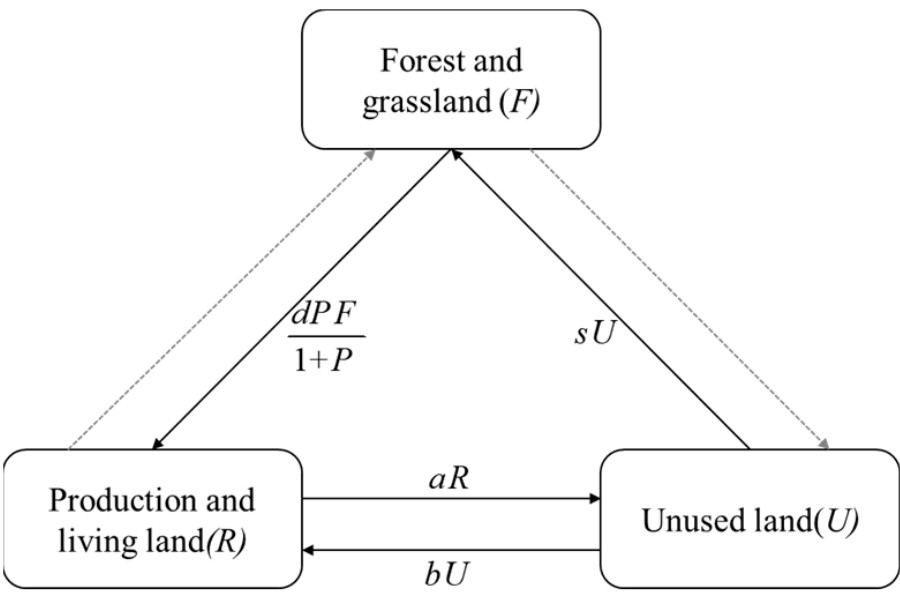

**Figure 2.** Evolution between different land-use types.

Since the unit of land-use-type area is not consistent with that of the population, the data are transformed into dimensionless data after normalization in the process of model construction and analysis. Through analyzing the data in Yuxi City over the years, it can be seen that the transformation function of population affecting land use is a nonlinear function, and the following function can be obtained through fitting:

$$f(t, P, F) = \frac{dPF}{1 + P} \tag{1}$$

where $P$ and $F$ are the population density and forest and grass land area at time $t$, respectively, and $d$ is the average reclamation or development capacity of the land. This function is generally interpreted as a Holling-II functional response function in mathematical definitions. Assuming that population growth conforms to the Logistic Retarded Growth model:

$$\frac{dP}{dt} = rP\left(1 + \frac{P(t)}{P_{\max}}\right) \tag{2}$$

where $r$ is the inherent growth rate of the population, and $P_{max}$ (greater than 0) is the maximum population that the environment can carry.

Based on the transformation mechanism mentioned above, in the interval $(t, t + \Delta t)$, the forest and grass land, production and living land and unused land change with time $t$, and with the help of the population retardation growth model, the population also changes with time $t$ under the constraints of the production and living land. Due to the need for both survival and population growth, more food is needed during population growth than during saturation, that is, the inherent growth rate of the population is a function of a time delay $t$-$\tau$. Considering the coupling relationship between the population, forest and grass land, the production and living land, and unused land comprehensively, and only discussing the impact of population development on the time delay, the following fractional time delay human–land coupling dynamics model with a Holling-II functional response function can be obtained [30]:

$$\begin{cases} D^\phi F(t) = \frac{-dF(t)P(t-\tau)}{1 + P(t-\tau)} + sU(t) \\ D^\phi R(t) = \frac{dF(t)P(t-\tau)}{1 + P(t-\tau)} - aR(t) + bU(t) \\ D^\phi U(t) = aR(t) - sU(t) - bU(t) \\ D^\phi P(t) = rP(t)\left[1 - \frac{h}{R(t)}P(t-\tau)\right] \end{cases} \tag{3}$$

where $\phi \in [0, 1]$ is the fractional order, and $r$ is the inherent growth rate of the population.

## 3. Results

### 3.1. Identification of Human–Land Coupling System Equilibrium Point

In order to make the application of the human–land coupling dynamics model meaningful in Yuxi City, it is necessary to find the sufficient conditions for the model's stability and discuss the stability of the model's equilibrium point and the sustainability of the model. Therefore, the model's equilibrium point is solved first, and the model's stability is further analyzed through the model's equilibrium point. When solving the fractional model, the Adama–Bashforth–Moulton predictive correction algorithm is applied [46]. To facilitate calculation, let the step size $\Delta t = 0.01$, $\phi = 0.9$, substitute $r = 0.0048$ into the model, and then through data analysis and simulation, parameters $a = 0.034$, $b = 0.012$, $s = 0.004$, $h = 0.05$, $d = 0.08$ can be obtained. After a normalized processing of the data for various types of areas and the population of Yuxi City in 1995, the initial values are $F(0)$, $R(0)$, $U(0)$, $P(0) = (0.07, 0.02, 0.4, 0.75)$; substitute them into Equation (3). Then, the equation

$$\begin{cases} 0 = \dfrac{-dF(t)P(t-\tau)}{1+P(t-\tau)} + sU(t) \\ 0 = \dfrac{dF(t)P(t-\tau)}{1+P(t-\tau)} - aR(t) + bU(t) \\ 0 = aR(t) - sU(t) - bU(t) \\ 0 = rP(t)\left(1 - \dfrac{h}{R(t)}P(t-\tau)\right) \end{cases}$$

is obtained, and the solution to this equation is

$$\begin{cases} F^* = 0.075617 \\ R^* = 0.021169 \\ U^* = 0.44984 \\ P^* = 0.423379 \end{cases}.$$

That is, the equilibrium point of the model is (0.075617, 0.021169, 0.44984, 0.423379), and the results are all positive. Therefore, it is consistent with the non-negative situation of land and population, which is systematic and meaningful and reflects the rationality of the model.

### 3.2. Visual Output and Expression of Human–Land Coupling Relationship

The basic reproduction number $R_0$ of the human–land coupling dynamics model is a very important parameter; it is said that in the state of balance, the amount of the increase in population brought by land-use changes is a sign that decides whether the land-use type changes or not, namely, only when $R_0 > 1$ does land-use-type transformation occur. If $R_0 < 1$, the transformation will tend to zero. Therefore, according to the calculation method of the basic regeneration number [47], the basic regeneration number of the model can be obtained after solving for the equilibrium point, $R_0 = \frac{bd+ds}{ash} = 18.8235 > 1$, indicating that the land-use types' transformation is significant in the model. Through calculation, $\omega_0 = 0.002358$, $\tau_0 = 728.403$ [30]. An arbitrary $\tau$ value which is less than $\tau_0$ was chosen arbitrarily, and MATLAB software was used for numerical simulation. Let $\tau = 700$, and it can be seen through numerical simulation that the human–land change trend over time and the three-dimensional evolution of the human–land relationship can be obtained after a period of damped oscillation (Figures 3 and 4). As $\tau = 700 < \tau_0$, both the theoretical and numerical simulation show that the equilibrium point (0.075617, 0.021169, 0.44984, 0.423379) is locally asymptotically stable, that is, under the influence of the population, the development and change value of all kinds of land use fluctuates, but eventually tends to be stable around the equilibrium point of the model.

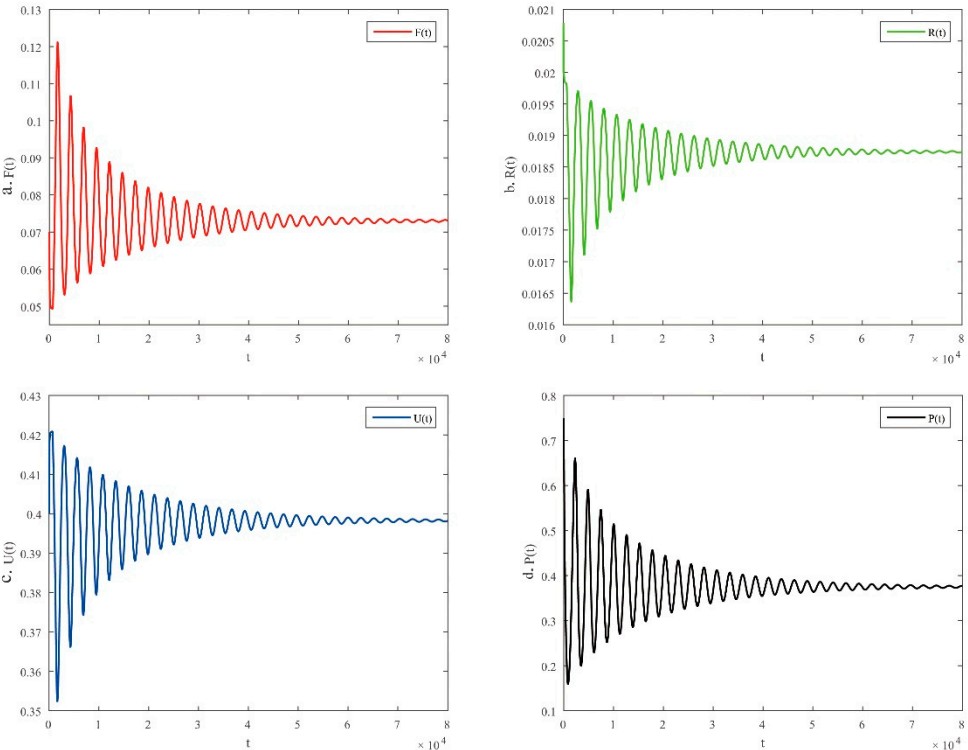

**Figure 3.** Evolution trend of people and land over time in Yuxi City when $\tau = 700$ ((**a**) change trends of forest and grass land over time; (**b**) change trends of production and living land over time; (**c**) change trends of unused land over time; (**d**) change trends of population over time).

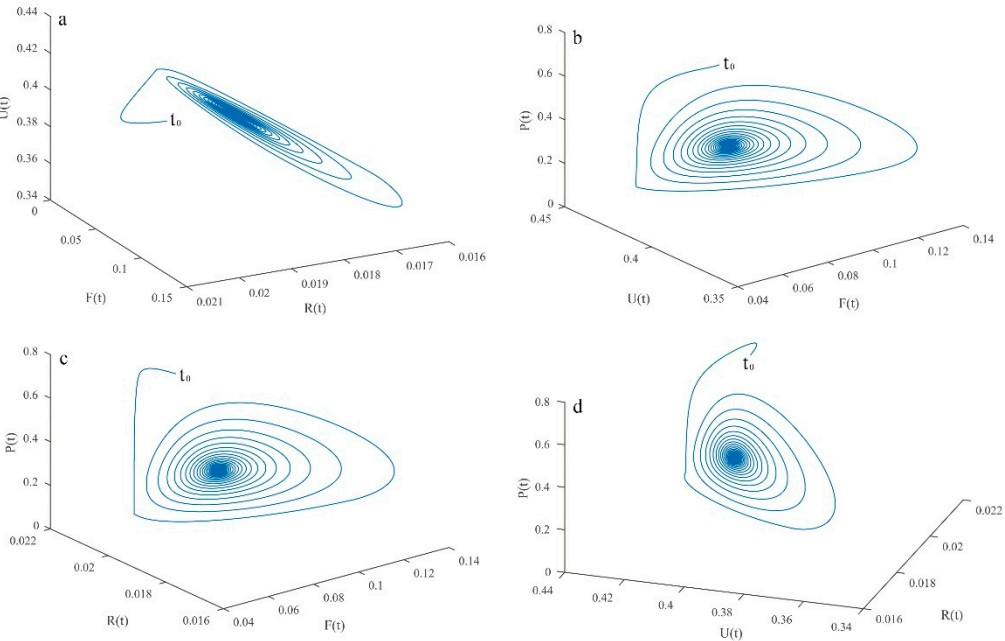

**Figure 4.** Three-dimensional map of human–land evolution in Yuxi City when $\tau = 700$ ((**a**) three-dimensional evolution of the unused land, forest and grass land and production and living land over time; (**b**) three-dimensional evolution of the population, unused land, and forest and grass land over time; (**c**) three-dimensional evolution of the population, production and living land and forest and grass land over time; (**d**) three-dimensional evolution of the population, unused land and production and living land over time).

As long as the time delay $\tau$ does not exceed $\tau_0 = 728.403$, after a $t = 8 \times 10^4$ simulation running time, the land-type area and population in Yuxi City will reach a balance point with time, and the system tends to be in a dynamic stable state. As can be seen from Figure 3a, the forest and grass land in the study area keeps changing with time, increasing and decreasing, but eventually tends to a stable value (0.075617). Similarly, under the system's equilibrium state, the production and living land tends to 0.021169 (Figure 3b), and the unused land tends to 0.44984 (Figure 3c). Under the three land-type transformation and constraint conditions, the population also changes with time, and the final population tends to 0.423379 (Figure 3d). Figure 4 shows the human–land three-dimensional evolutionary relationship at $\tau = 700$, which explains the dynamic characteristics of the human–land coupling system in a stable state. It can be seen that the system is in a stable state when the time delay is less than $\tau_0$.

Similarly, a value of $\tau$ that is larger than $\tau_0$ was arbitrarily selected for numerical simulation with MATLAB software. Let $\tau = 740$, and after a period of oscillation, the trend of human–land changes over time and the three-dimensional evolution of the human–land relationship is obtained (Figures 5 and 6). It can be seen that when $\tau = 740 > \tau_0 = 728.403$, the equilibrium point (0.075617, 0.021169, 0.44984, 0.423379) is no longer stable, that is, under the influence of population, the development of different land types fluctuates greatly at first; however, after the time delay is greater than a certain value, the values of all land-use types will tend to oscillate within a certain period, that is, Hopf bifurcation occurs. At this time, the area of the forest and grass land, the area of the production and living land and the area of the unused land will all show a periodic decline and increase.

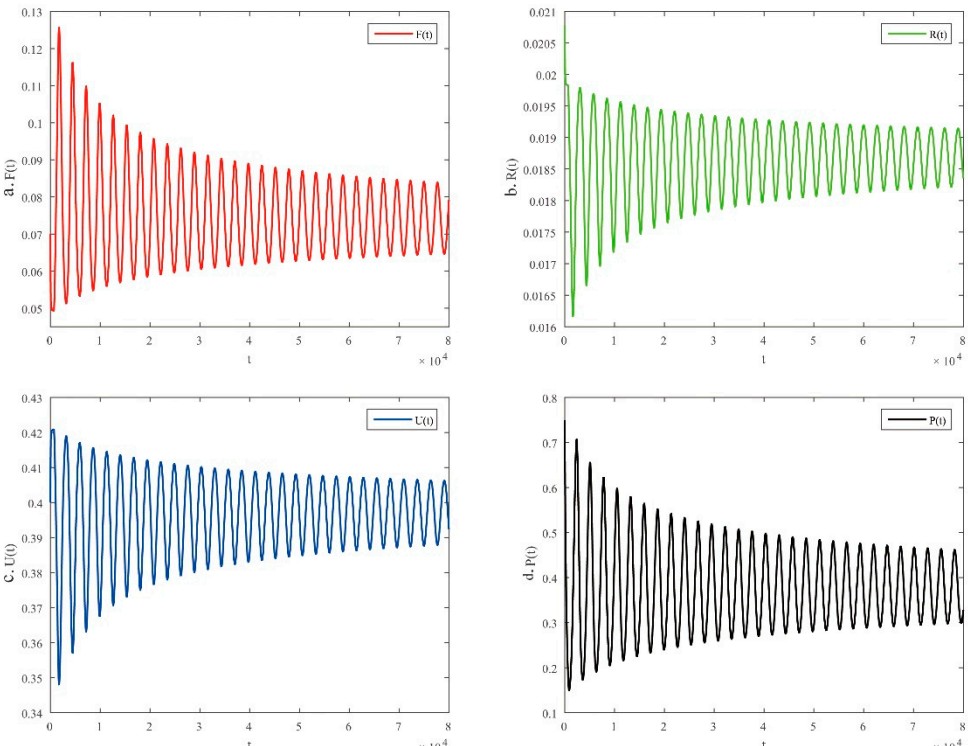

**Figure 5.** Evolution trend of people and land over time in Yuxi City when $\tau = 740$ ((**a**) change trends of forest and grass land over time; (**b**) change trends of production and living land over time; (**c**) change trends of unused land over time; (**d**) change trends of population over time).

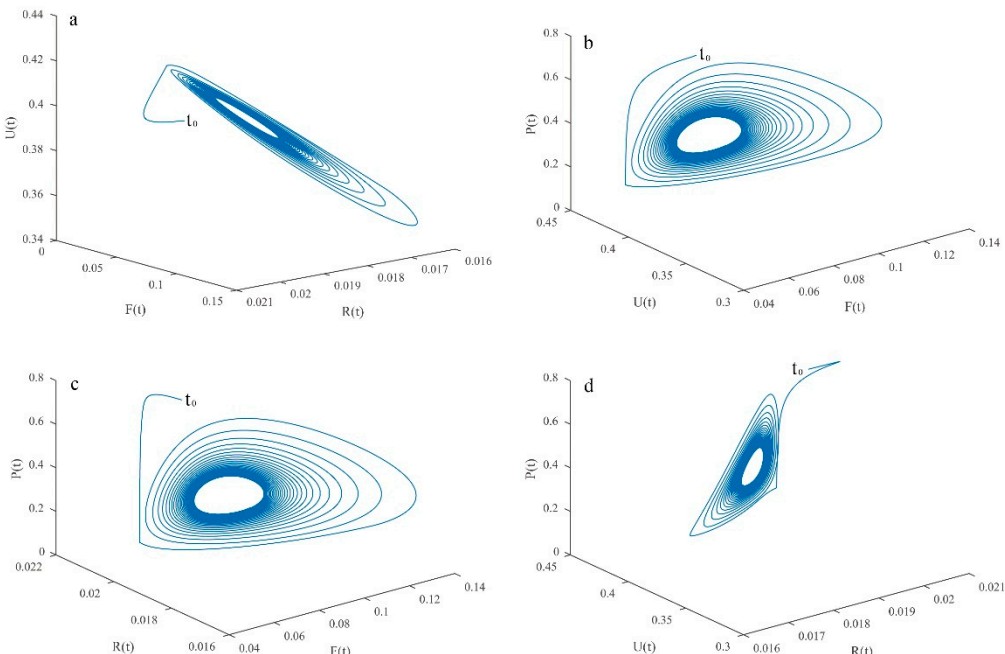

**Figure 6.** Three-dimensional map of human–land evolution in Yuxi City when τ = 740 ((**a**) three-dimensional evolution of the unused land, forest and grass land and production and living land over time; (**b**) three-dimensional evolution of the population, unused land, and forest and grass land over time; (**c**) three-dimensional evolution of the population, production and living land and forest and grass land over time; (**d**) three-dimensional evolution of the population, unused land and production and living land over time).

When the time delay τ exceeds $\tau_0$ = 728.403, after the simulation running time of $t = 8 \times 10^4$, the population and land types change from the initial large fluctuations to periodic changes around the equilibrium points (0.075617, 0.021169, 0.44984, 0.423379) (Figure 5); that is, in an unstable state, it is significantly different from the situation when the time delay τ is less than $\tau_0$ (Figure 3). When the time delay becomes larger, the area of forest and grass land $F(t)$, the area of production and living land $R(t)$ and the area of unused land $U(t)$ will change periodically. The forest and grass land change periodically around 0.075617 (Figure 5a), the production and living land change periodically around 0.021169 (Figure 5b), and the unused land changes periodically around 0.44984 (Figure 5c), and will not tend to a stable value. Figure 6 shows the three-dimensional evolution of the human–land relationship when τ = 740, which explains the dynamic characteristics of the human–land coupled system in the unstable state. It can be seen that when the time delay is greater than $\tau_0$, the system exhibits an unstable periodic oscillation phenomenon.

Comparing Figure 4 with Figure 6, we could see that when τ > $\tau_0$, the human–land evolution trend is more consistent with the actual situation, that is, with the periodic change in the population, the area of land types changes periodically. However, in order to tend to a stable state at τ < $\tau_0$, that is, the equilibrium state of the human–land system, relevant policies can be formulated and implemented by the government, that is, parameters can be controlled and adjusted.

### 3.3. Coupling Spatiotemporal Parameters of Mountain–Basin Human–Land Relationship

With the migration of the population from mountainous areas to basin areas, the agglomeration of the population leads to the occupation of agricultural land and the expansion of urban land [45,48]. Early on, due to the limited population, the urbanization level is not high, and coupled with the influence of the policy constrains, the migration of the population is also rare. The main way of life is farming, and the ecological and environmental effects caused by population migration are not obvious. However, with the

rapid improvement in the urbanization level and the acceleration of the migration of the population from rural to urban areas, land-use changes are accelerating. In mountainous areas, the arable land and construction land have been abandoned [49] and turned into a wasteland, that is, unused land. However, the wasteland will naturally recover into forest and grass land after a certain period of time. In basin areas, with the migration of the population, the demand for land is increasing, and the unused land will be gradually transformed into forest and grass land and farmland and construction land.

With the rapid increase in the population of the basin area, the land's resource-carrying capacity, industrial-supporting capacity and infrastructure are facing more challenges, driving the changes in the land-use pattern to meet the needs of population agglomeration. At the same time, the population in mountainous areas is decreasing, farmland and construction land will be abandoned, and ecological land has been restored. According to Model (3), two subsystems of mountainous areas and basin areas are distinguished, and human–land coupling evolution and development models of mountainous areas and basin areas are constructed, respectively:

$$
\begin{cases}
D^\phi F_M(t) = \frac{-d_M F_M(t) P_M(t-\tau_M)}{1+P_1(t-\tau_1)} + s_M U_M(t) \\
D^\phi R_M(t) = \frac{d_M F_M(t) P_M(t-\tau_M)}{1+P_M(t-\tau_M)} - a_M R_M(t) + b_M U_M(t) \\
D^\phi U_M(t) = a_M R_M(t) - s_M U_M(t) - b_M U_M(t) \\
D^\phi P_M(t) = r_M P_M(t)\left[1 - \frac{h_M}{R_M(t)} P_M(t-\tau_M)\right]
\end{cases}
\tag{4}
$$

$$
\begin{cases}
D^\phi F_B(t) = \frac{-d_B F_B(t) P_B(t-\tau_B)}{1+P_B(t-\tau_B)} + s_B U_B(t) \\
D^\phi R_B(t) = \frac{d_B F_B(t) P_B(t-\tau_B)}{1+P_B(t-\tau_B)} - a_B R_B(t) + b_B U_B(t) \\
D^\phi U_B(t) = a_B R_B(t) - s_B U_B(t) - b_B U_B(t) \\
D^\phi P_B(t) = r_B P_B(t)\left[1 - \frac{h_B}{R_B(t)} P_B(t-\tau_B)\right]
\end{cases}
\tag{5}
$$

In the formula, $\phi \in [0, 1]$ is the fractional order, and the relevant parameters are as follows: $F_M(t)$, $R_M(t)$, $U_M(t)$ and $F_B(t)$, $R_B(t)$, $U_B(t)$ respectively represent the forest and grass land, production and living land, unused land in mountainous areas and basin areas, and $F_M(t) + F_B(t) = F(t)$, $R_M(t) + R_B(t) = R(t)$, $U_M(t) + U_B(t) = U(t)$, $N = F(t) + R(t) + U(t) = 1$. $F_M(t)$, $R_M(t)$, $U_M(t)$ and $F_B(t)$, $R_B(t)$, $U_B(t)$ are the population density of mountainous areas and basin areas at time $t$, respectively. The production and living land [$R_M(t)$ and $R_B(t)$] will change into unused land [$U_M(t)$ and $U_B(t)$] during the period of $1/a_M$ and $1/a_B$, and then become forest and grass land [$F_M(t)$ and $F_B(t)$] through natural succession or ecological restoration after the interval of $1/s_M$ or $1/s_B$. The unused land can also be reclaimed or developed into production and living land after an interval $1/b_M$ and $1/b_B$. The average reclamation capacity of mountainous areas and basin areas is described by the constants $d_M$ and $d_B$ respectively. $r_M$ and $r_B$ are the natural growth rate of the population in mountainous areas and basin areas, respectively.

MATLAB software was used to further fit the land-type area and population for different periods of mountainous areas and basin areas, and the parameters of the human–land coupling dynamics model in mountainous areas and the human–land coupling dynamics model in basin areas were obtained, respectively (Table 2).

According to the existing data analysis, the conversion time parameters for production and living land to unused land of mountainous areas and basin areas in Yuxi City are $a_M = 0.0486$ and $a_B = 0.0126$, respectively, indicating that it takes about 20 years for production and living land in mountainous areas to be abandoned and then converted to unused land, while it takes about 80 years for basin areas. $a_M$ is greater than $a_B$, indicating that the conversion time of productive and living land to unused land in the basin area is slower than that in the mountainous area, mainly because the population growth in the basin area is faster than that in the mountainous area, and the demand for productive and living land for economic construction is larger. This trend will continue for a long time in

the future, making the conversion time of productive and living land to unused land much longer than that in the mountainous area.

**Table 2.** Fitting parameters of human–land coupling dynamics model in mountainous areas and basin areas of Yuxi City.

| Description | Parameters in Mountainous Areas | Parameters in Basin Areas |
|---|---|---|
| Conversion of production and living land into unused land | $a_M = 0.0486, 1/a_M = 20.5761$ | $a_B = 0.0126, 1/a_B = 79.3651$ |
| Conversion of unused land into production and living land | $b_M = 0.0062, 1/b_M = 161.2903$ | $b_B = 0.0139, 1/b_B = 71.9425$ |
| Conversion of unused land into forest and grass land | $s_M = 0.0051, 1/s_B = 196.0784$ | $s_B = 0.0028, 1/s_B = 357.1429$ |
| Land area required to maintain the unit individual | $h_M = 0.0335$ | $h_B = 0.0165$ |
| Average reclamation capacity | $d_M = 0.03$ | $d_B = 0.05$ |
| Natural growth rate of population | $r_M = 0.0563$ | $r_B = 0.151$ |

The time parameters of conversion from unused land to productive and living land in mountainous areas and basin areas are $b_M = 0.0062$ and $b_B = 0.0139$, respectively, indicating that unused land in mountainous areas will be reclaimed or developed into productive and living land again after about 160 years, while the time of conversion in basin areas is shorter, about 70 years. $b_M$ is less than $b_B$, indicating that the conversion time of unused land to productive and living land in basin areas is faster than that in mountainous areas. With the rapid development of the society and economy and the growth of the population in the basin area, the demand for construction land is always on the rise. Under the background of the policy of vigorously protecting cultivated land, the demand for construction land is mainly solved by the transformation of forest and grass land and unused land, which allows the unused land in the basin area to be transformed quickly and in a shorter time than that in the mountainous area.

The conversion time parameters of unused land to forest and grass land for mountainous areas and basin areas are $s_M = 0.0051$ and $s_B = 0.0028$, respectively, indicating that it takes about 200 years for unused land to convert to forest and grass land by natural succession or ecological restoration in the mountainous area, while it takes about 350 years for the basin area. $s_M$ is greater than $s_B$, showing that the conversion time from unused land to forest and grass land in the basin area is longer than that in the mountainous area. The main reason is that the social and economic development of the basin area has a radiation effect on the population and economy in the mountainous area, which makes the mountainous area gradually decline. The production and living land, such as farmland and construction land, is transformed into unused land and further transformed into forest and grass land due to their abandonment and extensive management, and the transformation speed is fast.

The land area parameters required for individual maintenance are $h_M = 0.0335$ and $h_B = 0.0165$, respectively. $h_M$ is greater than $h_B$, indicating that the land area required for each unit in basin areas is smaller than that in mountainous areas. The main reason is that the land-use intensity in mountainous areas is significantly lower than that in basin areas, and the land yield rate is far lower than that in the basin area, so that the land area required by each unit is larger than that in the basin area. The average reclamation or development capacity in mountainous areas and basin areas are $d_M = 0.03$ and $d_B = 0.05$, respectively. $d_M$ is less than $d_B$. This shows that the average reclamation and development capacity of the basin area is higher than that of the mountainous area, mainly because the economic development level, investment capacity and natural conditions of the basin area are better than those of the mountainous area.

## 4. Discussion

Under mathematical and geostatistical semantics, the order in a fractional differential equation can not only affect the dynamic characteristics of the fractional differential model, but also advance or delay the occurrence of the stability of the fractional differential model [50,51]. Therefore, the stability of the model can be improved by adjusting the model parameters. The influence of the time delay on the fractional order model is also dual. On the one hand, the time delay can make the fractional order model lose stability and lead to bifurcation. On the other hand, under certain conditions, the stability of the fractional order model can be improved with an appropriate time delay, and the occurrence of bifurcation can be further delayed [43]. Under the existing conditions of Yuxi City, $\tau_0 = 728.403$ is the critical point between the stability and instability of a human–land coupling system. The state of the human–land coupling system includes a stable equilibrium state and periodic oscillation state. The time delay can be used to determine the two states and how to adjust from periodic change to a stable state, or how to adjust to a stable state when periodic change is presented. Then, the time delay can be changed by changing the parameters $a$, $b$, $s$, $h$, $d$ and $r$. The time delay is like an invisible hand, which can not only optimize the allocation of land resources, but also give a warning signal when the production and living land area tends to be unstable, reminding people to make the land-use evolution stay stable by regulating the average reclamation or development capacity of individuals ($d$) and the inherent growth rate of population changes ($r$). The rate by which the unused land is recultivated or developed ($b$) and an individual's average reclamation or development capacity ($d$) can be artificially controlled. When the human–land evolutionary system tends to be stable, $b \in [0.012, 1]$, $d \in [0.08, 1]$ and $d$ is greater than $b$, indicating that the average reclamation or development capacity has a greater impact on the evolutionary stability of land use than the transformation rate of unused land. The optimal state of balancing a human–land coupling system is the optimal state of sustainable development, which is an ideal state. By adjusting these parameters, the system can be as close to the ideal state as possible.

Chen [47] constructed an integer order land dynamics model with a time delay in 2017, simulated the data of population and land-use change over the years in China, and found that the service life of subsistence land in China is about 25–70 years ($a = 0.04$), the time for reclaiming or developing the wasteland into subsistence land is about 100 years ($b = 0.011$), and the time for restoring the wasteland to original land is about 1000 years ($s = 0.005$). By comparison, there are similarities and differences with the parameters in this study, which are highlighted by the conversion time parameter $s$ of unused land to forest and grass land. Theoretically, it will take a long time for the degraded land to be restored to the original forest and grass land through natural succession, but with the progress of science and technology, this time will be greatly shortened, especially in the mountainous area of Yuxi City with its better ecological environment. Of course, these time parameters are obtained from existing data analysis. The reclamation or development rate of unused land ($b$) and the average reclamation or development capacity ($d$) are both controllable factors, and the average reclamation or development capacity has a greater impact on the stability of human–land systems.

This study based on systems integration thought that by analyzing the relationship between population and land based on coupled differential equations of forest and grass land, production and living land, unused land and population, it could build a fractional order human–land coupling dynamics mode with a Holling-II-type land conversion rate and time delay. The human–land coupling mechanism of the mountain–basin system is quantitatively described, and a new simulation direction is provided for coordination and optimization. However, the application of the method still needs to be improved in the future. First, since the human–land coupling relationship is a long-term process, the accuracy of parameter estimations depends on long-term sequence data. However, the existing collected data are only 24 years old, which is too small compared to the land conversion cycle. The model built is a high-dimensional and nonlinear differential system, and the

estimation of its parameters is inherently difficult. In addition, the lack of data makes it more difficult to achieve accurate model parameters such as for population growth rate and land conversion rate, etc. Secondly, in the classification process, there are problems such as the inaccuracy of various land types. For example, forest and grass land are assumed as primitive land in the model, but in fact, the existing forest and grass land have been transformed by human beings for a long time and lost their original nature. Finally, the coupling relationship between humans and land is complex. Although population is the main influencing factor for land-use changes, there are still more disturbance variables. The differential effects of special regional policies (such as urbanization in nearby areas, relocation of poor people from inhospitable areas, rural revitalization, development of plateau agriculture, ecological protection, etc.), technical means and micro-farmer behaviors on the coupled evolution of mountain–basin human–land systems should also be considered. Therefore, how to further build a multi-factor land-use dynamics model for specific regions and explore the coupled evolution mechanism of mountain–basin human–land systems should be the future direction of efforts.

## 5. Conclusions

A human–land coupling system is a nonlinear system, which is a differential equation composed of several coupling factors. In this study, the coupling factors of forest and grass land, production and living land, unused land and population were considered to construct a fractional order dynamics model with a time delay based on a Holling-II transformation rate. The stability of the system and its regulation mechanisms are discussed based on the solution of the equilibrium point of the system. It is known that the coupling state of human–land systems includes a stable equilibrium state and periodic oscillation state, and the two states of the system can be determined according to the time delay. How to adjust from periodic change to a stable state, or how to adjust to a stable state when periodic change is presented, the time delay can be changed by changing the parameters of the model.

On the basis of constructing the human–land coupling evolution model of Yuxi City, the human–land coupling evolution and development models of mountainous areas and basin areas were respectively constructed based on the internal relationship between the mountainous area and the basin area. The parameters of the human–land coupling evolution and development models were obtained by simulation analysis, using the existing data. According to the analysis of the parameters, the mountainous area and the basin area playing different roles in the process of human–land evolution systems tend to a stable state, i.e., an equilibrium state, and there are significant differences in the transformation time among different land types. Therefore, different land-use regulation strategies should be selected in different regions. In mountainous areas, the average reclamation or development capacity should be reduced to lower the conversion of forest and grass land to production and living land, and at the same time, excessive interference of human beings should be reduced to speed up the natural recovery and succession of unused land to forest and grass land. In basin areas, in addition to reducing the average reclamation or the development capacity, the reclamation or development rate of the idle land and degraded land should be increased, and the conversion of idle land into productive and living land should be encouraged with the help of certain scientific and technological means.

**Author Contributions:** Conceptualization, L.W.; methodology, L.W.; software, L.W.; validation, L.W. and B.X.; formal analysis, L.W.; investigation, L.W.; resources, L.W.; data curation, L.W.; writing—original draft preparation, L.W.; writing—review and editing, L.W., Y.Y. and B.X.; supervision, B.X.; funding acquisition, L.W. All authors have read and agreed to the published version of the manuscript.

**Funding:** This research was funded by the National Natural Science Foundation of China, grant number 42161041.

**Institutional Review Board Statement:** Not applicable.

**Informed Consent Statement:** Not applicable.

**Data Availability Statement:** Not applicable.

**Conflicts of Interest:** The authors declare no conflict of interest.

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
