# Peer review of "Modeling Analysis on Coupling Mechanisms of Mountain–Basin Human–Land Systems: Take Yuxi City as an Example"

_land, doi:10.3390/land11071068_

Round 1

Reviewer 1 Report

I think the idea behind the article is good, but the overall quality of presentation is flawed. I would suggest a focus on data presentation, language usage and literature review.

In details, 1) literature review needs a strong reinforcement, considering publications out of the studied country.

2) originality and novelty of the approach are not discussed in details.

3) we would see a better definition of coupling/decoupling processes. Is it an economic process? Is it a social or environmental process? Please clarify and provide better definitions, supported by relevant literature.

Thank you. 

Reviewer 2 Report

Overall, the paper is well written and presented. However, more could be done. Please find below my comments/suggestions that could improve the article

-       The authors should include the case study in the title.

-       In the abstract and in the introduction the authors should clearly indicate the objectives of this study.

-       Introduction: Please, clarify what are the innovative contributions of this study to science. Highlight the aspects in which your research departs from the existing literature.

-       Study area: case study contextualization is missing. All elements that could help the reader better picture what looks like the area we are talking about. I believe one of the aims of the work is to make readers understand the magnitude of the issue authors are debating. In addition, the authors should be clear about why choosing this particular region in China.

-       Methodology: it needs more explanation of issues and alternative approaches to overcome.

-       Please include a methodological framework.

-       Could you explain the weaknesses of this research/method? Limitations?

-       Discussion: it is fairly written, but I will suggest homing in on how the methods and outcome could be applicable elsewhere. This will help boost the potential international readership of the manuscript. 

-       The conclusions section should be expanded. More specifically, please expand on how this study will be addressed as a part of future research.

-       Minor grammar and punctuation errors can be found throughout the text and need to be corrected.

Reviewer 3 Report

The reviewed manuscript analyse the coupling mechanism of the mountain-basin human-land system in Yuxi City, Yunnan province. The submitted manuscript fits the scope of the journal. However, major revision before publication is needed. The methodology is unclear: there are missing research questions or hypotheses, and used datasets and methods are not appropriately elaborated. The formulas used are difficult to follow because a lot of valuation factors are abbreviated using only one letter. The results and discussion sections must be improved, there are only facts stated without explaining how the results could be used to improve the current land management procedure. The results should be compared to the results from similar studies, cited in the references. The discussion section should include the intended use of the proposed methodology, indicating how this knowledge should be used to make better decisions concerning physical planning projects. The manuscript needs at least a few maps – a general overview, thematic maps and so on... There is no single map in the manuscript and it is discussing the land-human interaction.

Round 2

Reviewer 1 Report

Good and standard paper

Reviewer 2 Report

The authors answered my comments from the last review. Therefore, in my opinion, the manuscript can be accepted for publication.

Reviewer 3 Report

I believe the manuscript has been sufficiently improved to warrant publication in Land.